

# Development of polymorphic EST-SSR markers and characterization of the autotetraploid genome of sainfoin (*Onobrychis viciifolia*)

Shuheng Shen*, Xutian Chai*, Qiang Zhou, Dong Luo, Yanrong Wang and Zhipeng Liu

State Key Laboratory of Grassland Agro-ecosystems, Key Laboratory of Grassland Livestock Industry Innovation, Ministry of Agriculture and Rural Affairs, College of Pastoral Agriculture Science and Technology, Lanzhou University, Lanzhou University, Lanzhou, Gansu, China

* These authors contributed equally to this work.

## ABSTRACT

**Background**. Sainfoin (*Onobrychis viciifolia*) is a highly nutritious, tannin-containing, and tetraploid forage legume. Due to the lack of detailed transcriptomic and genomic information on this species, genetic and breeding projects for sainfoin improvement have been significantly hindered.

**Methods**. In this study, a total of 24,630,711 clean reads were generated from 14 different sainfoin tissues using Illumina paired-end sequencing technology and deposited in the NCBI SRA database (SRX3763386). From these clean reads, 77,764 unigene sequences were obtained and 6,752 EST-SSRs were identified using *de novo* assembly. A total of 2,469 primer pairs were designed, and 200 primer pairs were randomly selected to analyze the polymorphism in five sainfoin wild accessions.

**Results**. Further analysis of 40 sainfoin individuals from the five wild populations using 61 EST-SSR loci showed that the number of alleles per locus ranged from 4 to 15, and the expected heterozygosity varied from 0.55 to 0.91. Additionally, by counting the EST-SSR band number and sequencing the three or four bands in one sainfoin individual, sainfoin was confirmed to be autotetraploid. This finding provides a high level of information about this plant.

**Discussion**. Through this study, 61 EST-SSR markers were successfully developed and shown to be useful for genetic studies and investigations of population genetic structures and variabilities among different sainfoin accessions.

## INTRODUCTION

Sainfoin (*Onobrychis viciaefolia*) is a cross-pollinated, autotetraploid and perennial legume ($2n = 4x = 28$) that is commonly used as a silage. The nutritional value of sainfoin is universally recognized, and it is known to be rich in proteins and secondary metabolites. Sainfoin can also fix atmospheric nitrogen through its symbiotic relationship with rhizobia. The origin center of sainfoin is known as the Middle East and Central Asia. In China,

Corresponding author
Zhipeng Liu, lzp@lzu.edu.cn

sainfoin is mainly grown in the northeast, north and northwest regions, including Gansu province. It does not cause bloat in grazing animals, and can provide palatable, high-quality forage (*Bhattarai, Coulman & Biligetu, 2016*; *Frame, 2005*). Sainfoin contains high levels of condensed tannins that sainfoin has shown to reduce parasites in the ruminant digestive tract and provide environmental benefits by reducing methane emissions from ruminant animals (*Malisch et al., 2015*; *Sottie et al., 2014*; *Bhattarai et al., 2018*).

In recent years, there was a renewed interest in sainfoin for its use in animal diets. Several studies indicated that the voluntary intake of sainfoin by grazing heifers is higher than alfalfa (*Medicago sativa*) (*Parker & Moss, 1981*; *Kempf et al., 2016*). *Scharenberg et al. (2007)* reported that sainfoin was more palatable than birdsfoot trefoil when given to sheep. However, sainfoin is a relatively understudied forage legume compared to alfalfa or clovers (*Trifolium* spp.). Therefore, exploitation and conservation of sainfoin germplasms became important. Also, knowledge of sainfoin genetic diversity and structures has become a prerequisite for successful sainfoin conservation programs (*Sun, Salomon & Bothmer, 2002*). To date, reports on sainfoin transcriptomes and genomics are very limited, and this hinders many genetic and breeding projects for this plant.

Simple sequence repeats (SSR) for microsatellite markers are tandem repeated mono-, di-, tri-, tetra-, penta- or hexa-nucleotide sequences that possess high information content, co-dominance and locus specificities and are easier to be detected compared to other molecular markers. SSR markers were successfully used to study genetic variation, genetic mapping, and molecular breeding for many plants (*Naghavi et al., 2007*; *Gupta, Langridge & Mir, 2010*; *Salem et al., 2010*; *Prasanna et al., 2010*; *Li et al., 2011*). Compared to genomic-SSRs, expressed sequence tag (EST) EST-SSRs were reported to provide higher levels of transferability across the related species, because EST-SSR markers were identified in the coding regions of the genome and the identified sequences are more conserved among homologous genes (*Wu et al., 2014*). EST-SSR markers have now been developed for many plant species using Illumina sequencing technologies. These plants include alfalfa (*Liu et al., 2013a*), wheat (*Gupta & Varshney, 2000*), adzuki bean (*Kang et al., 2015*), edible pea (*Nisar et al., 2017*), mung bean (*Chen et al., 2015*), and Siberian wildrye (*Zhou et al., 2016*). Current studies on sainfoin genetic diversity, map-based cloning, and molecular breeding lag behind many legume crops due mainly to the lack of genomic information. Only 101 polymorphic EST-SSRs were confirmed by individual sainfoin plants (*Kempf et al., 2016*; *Mora-Ortiz et al., 2016*). The current available EST-SSR primers are not sufficient for the studies on sainfoin genetic diversity, fingerprinting, and genetic mapping. These limitations have hindered the molecular breeding for sainfoin yield and nutritional value improvements.

Recent studies showed that next-generation transcriptome sequencing and Roche/454 genome sequencing technologies are effective solutions for generating large-scale genomic information in short periods of time and at reasonable costs, even for non-model plant species (*Wang et al., 2010*). Because these sequencing technologies also allow extensive investigations on alternative RNA splicing, discovery of novel transcripts, and identifications of gene boundaries at the single-nucleotide resolution level, massive parallel transcriptome sequencing has provided great opportunities to revolutionize studies of

plant transcriptomics. For example, EST-SSR markers can now be quickly developed using a bioinformatic data mining approach. Because EST-SSR markers have many advantages over genomic SSR markers during marker development, we decided to analyze the complex tetraploid sainfoin genome to develop useful EST-SSR markers for future studies. Compared with SNP markers, EST-SSRs are multi-allelic and it have a higher level of polymorphism and transferability across related species. These features make EST-SSR markers highly variable and useful for distinguishing closely related populations or varieties compared to genomic SSR markers. EST-SSR markers are known to be easily accessible, present in gene-rich regions, associated with transcription, useful for candidate gene identification, and transferrable between closely related species (*Thiel et al., 2003*). We considered that the EST-SSR markers developed for sainfoin using an RNA-seq technology should benefit sainfoin improvement projects, such as genetic diversity analysis, germplasm identification, comparative genetics, phylogenetic relationship, QTL analysis, linkage mapping and marker-assisted selection.

In this study, our aim was to use transcriptome sequencing of 14 sainfoin tissues on the Illumina Hiseq2500 sequencing platform. The objective of this study was to achieve a valuable sequence resource and develop some high polymorphism EST-SSR markers that would allow a better understanding of the genetic diversity of sainfoin. By counting the EST-SSR band number and sequencing the bands in sainfoin individuals, we aimed to re-verify sainfoin autotetraploidy.

## MATERIALS & METHODS

### Tissue sampling and total RNA isolation

Sainfoin callus cells, emerging tidbits (<2 cm), young inflorescences (2–4 cm), inflorescences (4–6 cm), mature inflorescences (6–8 cm), developing seed pods (16 days after pollination (dap)), mature seed pods (24 dap), roots, germinated seeds (36 h after seed germination), young stems (less lignified), stems (moderately lignified), mature stems (highly lignified), young compound leaves, and mature compound leaves were harvested (Fig. 1). A total of 14 tissues in the Fig. 1, The callus cells were induced from inflorescences at 25 °C on solid MS medium containing 2,4-dichlorophenoxyacetic acid (3.0 mg/L) for 30 days under a 16/8 h (light/dark) light cycle (*Ma et al., 2012*). (A–G and J–N) were harvested from the same individual plant, these tissues used in this study were from the same 2-year-old plants grown inside a greenhouse set at 22 °C and a 16/8 h (light/dark) light cycle at Lanzhou University, radicle and germ (H and I) were harvested from three seedlings germinated from seeds. The sampled tissues were immediately frozen in liquid nitrogen and stored at −80 °C until use. Total RNA was isolated from 14 individually collected samples using the RNeasy Plant Mini Kit (Qiagen, Cat. #74904) as instructed. Concentrations of the RNA samples were determined using a NanoDrop ND1000 spectrophotometer (Thermo Scientific, Waltham, MA, USA).

### cDNA library construction and sequencing

To better elucidate tissue-specific RNA transcription, each RNA sample was adjusted to 400 ng/μL. Twenty micrograms of total RNA was taken from each RNA sample and

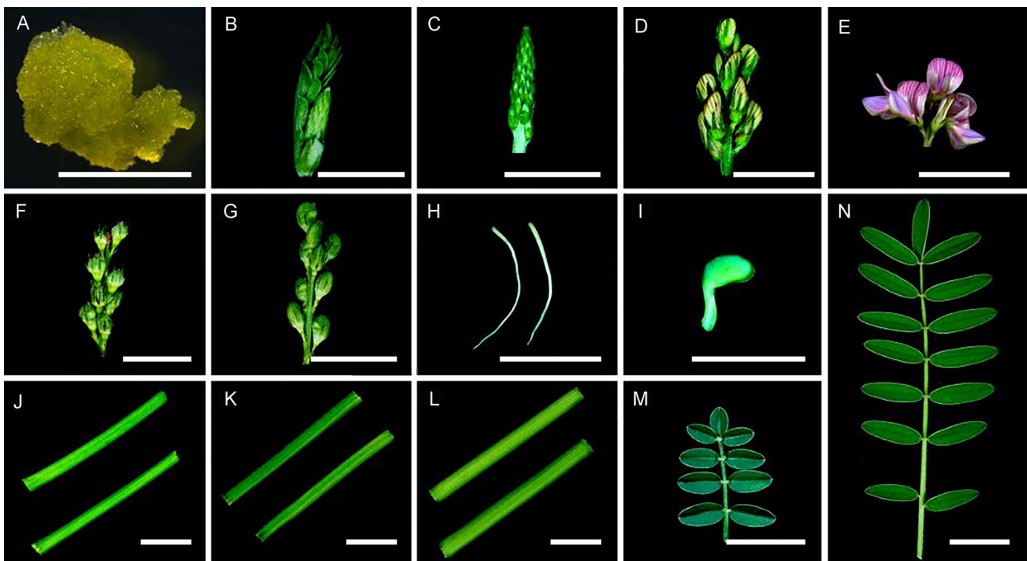

**Figure 1   Tissues used in this study. Samples were collected as described in the 'Materials and Methods' section.** (A) Callus cells. (B) An emerging tidbit. (C) A young inflorescence. (D) An inflorescence. (E) Mature inflorescence. (F) Developing seed pods. (G) Mature seed pods. (H) Roots. (I) A germinated seed. (J) Young stems. (K) Stems. (L) Mature stems. (M) A young compound leaf. (N) A mature compound leaf. Bar = 1 cm.

pooled prior to cDNA library preparation using the mRNA-Seq Sample Preparation Kit (Illumina Inc., Beijing, China). Briefly, poly (A) mRNA was isolated from the pooled total RNA sample using magnetic oligo dT beads and used to synthesize first-strand cDNA with random hexamer primers and reverse transcriptase (Invitrogen, Beijing, China). Short cDNA fragments were purified using a MinElute PCR Purification Kit (Qiagen, Beijing China), resuspended in an EB buffer (Qiagen), and poly A was added. Sequencing adapters were ligated to the short cDNA fragments, and the libraries were sequenced using the Illumina HiSeq2500 sequencing platform at the BioMarker Company (Beijing, China). Processing of fluorescent images for sequence base-calling and calculation of quality values were performed using the Illumina data processing pipeline, which yielded 100-bp paired-end reads.

### Sequence assembly and annotation

Before assembly, the raw reads were filtered to remove poly A/T, low-quality sequences, and empty reads or reads with more than 10% of bases having $Q < 30$. The assembly of a *de novo* transcriptome using clean reads was performed using the short-read assembling program Trinity. Contigs were generated after combining the reads with a certain degree of sequence overlap. Paired-end reads were used to detect contigs from the same transcript and the distances between contigs. Scaffolds were produced using N, representing different sequences between two contigs but connecting these two contigs together. Gaps between scaffolds were filled with paired-end reads and the reads with the lowest numbers of Ns. The resulting sequences were referred to as unigenes. The EST-SSR annotation positions

of these unigenes were determined using BLASTX alignment ($e$-value $< 10\text{-}5$) against the sequences in the databases: NR, Swiss-Prot, KEGG, COG, and unigene sequences. After NR annotation, unigene GO annotations were conducted using the Blast2GO algorithm. GO functional classifications of the unigenes were performed using the WEGO software.

## Detection of EST-SSR markers and designing of primers

EST-SSR markers were detected in the assembled unigenes using the Simple Sequence Repeat Identification Tool. The EST-SSRs were considered to contain mono-, di-, tri-, tetra-, penta-, and hexa-nucleotides with a minimum of ten, six, five, five, five, and five repeats, respectively. EST-SSR primers were designed using the BatchPrimer3 software and synthesized by the Shanghai Sangon Biological Engineering Technology (Shanghai, China).

## Sources of wild sainfoin populations

To produce results representing a wide range of sainfoin populations, we collected 40 individual wild sainfoin plants from five different locations (eight plants per location): Minqin, Jingyuan, Yuzhong, Huining and Maqu (Table S3). After air drying, leaves were taken from each plant and stored at room temperature until use.

## DNA extraction

DNA was extracted from the harvested and dried leaves using the Nucleon Phytopure Genomic DNA extraction kit (Ezyp column plant genomic DNA extraction kit, Sangon Biotech Shanghai, China) on the samples described above using a protocol reported previously (*Mora-Ortiz et al., 2016*). The quality of each isolated genomic DNA was examined using 0.8% agarose gel electrophoresis, and the concentration of each genomic DNA sample was determined using a NanoDrop 8000 spectrophotometer (Nanodrop Technologies Wilmington, DE). The DNA samples were diluted individually in TE buffer to 25 ng DNA/$\mu$L prior to PCR amplification.

## Amplification of EST-SSRs using polymerase chain reaction (PCR)

A total of 200 primer pairs were produced for this study and used to amplify EST-SSRs from the genomic DNA samples from the 40 wild sainfoin plants. PCR amplifications were performed in 5-$\mu$L reactions (0.5 $\mu$L of DNA, 2.5 uL of 2$\times$ mix (e.g., 0.5 $\mu$L of 2$\times$ PCR buffer, 1 $\mu$L of 1 mM dNTPs, 0.4 $\mu$L of 25 mM MgCl$_2$, 0.1 $\mu$L of Taq DNA polymerase), 0.5 $\mu$L (5 pmoL/$\mu$L) of forward and reverse primers, and 1uL of sterile distilled water). Three microliters was added to the PCR reaction in each tube, and PCR was performed using a PCR-100TM Thermal cycler set at 94 °C for 4 min, followed by 35 cycles of 94 °C for 1 min at a specific annealing temperature for 30 s and 72 °C for 20 s. The final extension was at 72 °C for 7 min. The resulting PCR products were resolved using 8.0% non-denaturing polyacrylamide gels (Lot# I20826, GelStain; Beijing TransGen Biotech. Co., Ltd., Beijing, China) after electrophoresis.

## Construction of T vectors

Fragments of 200–500 bp were PCR amplified from sainfoin genomic DNA using the gene-specific primers. The PCR products were purified and these fragments were then

cloned into pGEM-T Easy Vector (Promega, Madison, WI, USA). The constructs were transferred into *E.coli* DH5 $\alpha$ using the freezing/heat-shock method, and sequenced by Shanghai Sangon Biological Engineering Technology using Sanger dideoxy (Shanghai, China).

### Diversity analysis

The observed heterozygosity (Ho) was calculated as previously shown (*Liu, Liu & Yang, 2007*), and the corrected heterozygosity (He), corrected for sample size, and the average polymorphism information content (PIC) were analyzed using the ATETRA 1.2.a software program. Only specific bands that could be unambiguously scored across all individual plants were used in this study. A clustering analysis was used to generate a dendrogram using the unweighted pair-group method with arithmetic mean (UPGMA) and Nei's unbiased genetic distance with the NTSYSPC 2.0 software package. We used STRUCTURE 2.3.4 to generate a structure map.

## RESULTS

### RNA sequencing and *de novo* assembly

A cDNA library representing 14 different sainfoin tissues (Fig. 1) was sequenced, and a total of 26,912,927 raw reads were obtained (Table 1). After rigorous quality checks and data filtering, a total of 24,630,711 high-quality clean reads were obtained, and about 92% of them had quality (*Q*)-scores above Q30. These clean reads together contained a total of 6,264,706,761 nucleotides (nt), without N, and about 45% GC content. The high-quality reads were deposited in the U.S. National Center for Biotechnology Information (NCBI) sequence read archive (SRA) database (SRX3763386). From these high-quality reads, a total of 2,678,687 contigs with a mean length of 72.26 bp and an N50 length of 69 bp were generated using *de novo* assembly. The total number of unigenes from the paired-end reads was 77,764, and 27,437 of them had distinct clusters. A total of 50,327 unigenes had distinct singletons, and the total length of these unigenes was 53,035,704 bp. The average length of a unigene was 682.01 bp with an N50 value of 1,209 bp. Of the 77,764 unigenes, the length of 50,327 unigenes ranged from 200 to 500 bp, the length of 22,096 unigenes ranged from 500 to 2,000 bp, and the length of 5,341 unigenes was above 2,000 bp (Table 1). Also, 36,353 of the 77,764 unigenes were successfully annotated according to the NR, Pfam, Swiss-Prot, KEGG, COG, and GO databases (Table 2), and 10,387 unigenes were assigned to COG classifications.

### Frequency and distribution of EST-SSRs

A total of 6,752 potential EST-SSRs were identified in the 77,764 unigenes (Table 1) and used to design 2,469 primer pairs. Since 1,271 unigenes contained more than one EST-SSR, the types and distributions of the total 6,752 potential EST-SSRs were analyzed. As shown in Fig. S4, the density of single nucleotide repeats was the highest (88 of SSRs per Mb), followed by trinucleotide repeats (64 of SSRs per Mb). The most abundant repeat type was the mono-nucleotide repeat (2,906 repeats or 43.04% of the total repeats), followed by tri-nucleotide repeats (2,262, 33.50%), di-nucleotide repeats

**Table 1  Summary of the *de novo* assembled sainfoin EST-SSRs.**

| Category | Items | Number |
|---|---|---|
| Raw reads | Total raw reads | 26,912,927 |
| | Total clean reads | 24,630,711 |
| Clean reads | Total clean nucleotides (nt) | 6,264,706,761 |
| | Q30 percentage | 91.52% |
| | *N* percentage | 0% |
| | GC percentage | 44.98% |
| | Total number | 2,678,687 |
| Contigs | Total length (bp) | 193,558,725 |
| | Mean length (bp) | 72.26 |
| | N50 (bp) | 69 |
| | Total number | 77,764 |
| | Total length (bp) | 53,035,704 |
| Unigenes | Mean length (bp) | 682.01 |
| | N50 (bp) | 1,209 |
| | Distinct clusters | 27,437 |
| | Distinct singletons | 50,327 |
| | Total number of examined sequences | 77,764 |
| EST-SSRs | Total number of identified SSRs | 6,752 |
| | Number of SSR-containing sequences | 4,988 |
| | Number of sequences containing more than one SSR | 1,271 |

**Table 2  Functional annotation of sainfion transcriptome.**

| Category | Number | Percentage (%) |
|---|---|---|
| NR annotation | 35,421 | 46% |
| KOG Annotation | 19,555 | 25% |
| Pfam Annotation | 24,282 | 31% |
| Swiss-pro Annotation | 21,973 | 28% |
| KEGG annotation | 11,923 | 15% |
| COG annotation | 10,387 | 13% |
| GO annotation | 22,237 | 29% |
| All | 36,353 | 47% |

(1,287, 19.06%), quad-nucleotide repeats (263, 3.90%), hexa-nucleotide repeats (19, 0.28%), and penta-nucleotide repeats (15, 0.22%) (Table 3). Moreover, the EST-SSRs with five tandem repeats were the most common EST-SSRs (24.88%), followed by ten tandem repeats (22.07%), six tandem repeats (16.88%), eleven tandem repeats (10.19%), seven tandem repeats (7.09%), and the remaining tandem repeats (<5%) (Table 3). In addition, the GO enrichment for the 6,752 SSR-containing unigenes was done using the agriGO algorithm (http://systemsbiology.cau.edu.cn/agriGOv2/index.php) and the 971,445 unigenes available in the database as the reference. The results of the GO enrichment analysis indicated that the proportion of the "transcription"-related (GO:0003674) unigenes was significantly enriched (Fig. S5).

**Table 3** Length distribution of EST-SSRs determined by the number of nucleotide repeats.

| Number of repeats | Mono- | Di- | Tri- | Quad- | Penta- | Hexa- | Total | Percentage (%) |
|---|---|---|---|---|---|---|---|---|
| 5 | 0 | 0 | 1,430 | 231 | 10 | 9 | 1,680 | 24.88 |
| 6 | 0 | 474 | 626 | 31 | 4 | 5 | 1,140 | 16.88 |
| 7 | 0 | 287 | 191 | 0 | 1 | 0 | 479 | 7.09 |
| 8 | 0 | 204 | 15 | 1 | 0 | 1 | 221 | 3.27 |
| 9 | 0 | 138 | 0 | 0 | 0 | 3 | 141 | 2.09 |
| 10 | 1,370 | 120 | 0 | 0 | 0 | 0 | 1,490 | 22.07 |
| 11 | 627 | 60 | 0 | 0 | 0 | 1 | 688 | 10.19 |
| 12 | 270 | 4 | 0 | 0 | 0 | 0 | 274 | 4.06 |
| 13 | 218 | 0 | 0 | 0 | 0 | 0 | 218 | 3.23 |
| 14 | 152 | 0 | 0 | 0 | 0 | 0 | 152 | 2.25 |
| 15 | 112 | 0 | 0 | 0 | 0 | 0 | 112 | 1.66 |
| 16 | 76 | 0 | 0 | 0 | 0 | 0 | 76 | 1.13 |
| 17 | 41 | 0 | 0 | 0 | 0 | 0 | 41 | 0.61 |
| 18 | 11 | 0 | 0 | 0 | 0 | 0 | 11 | 0.16 |
| 19 | 8 | 0 | 0 | 0 | 0 | 0 | 8 | 0.12 |
| 20 | 2 | 0 | 0 | 0 | 0 | 0 | 2 | 0.03 |
| 21 | 9 | 0 | 0 | 0 | 0 | 0 | 9 | 0.13 |
| 22 | 5 | 0 | 0 | 0 | 0 | 0 | 5 | 0.07 |
| 23 | 4 | 0 | 0 | 0 | 0 | 0 | 4 | 0.06 |
| 24 | 1 | 0 | 0 | 0 | 0 | 0 | 1 | 0.01 |
| Total | 2,906 | 1,287 | 2,262 | 263 | 15 | 19 | 6,752 | – |
| Percentage (%) | 43.04 | 19.06 | 33.50 | 3.90 | 0.22 | 0.28 | – | – |

## Development of EST-SSR markers

Using the SSR-containing unigene sequences, 200 primer pairs were randomly chosen from the 2,469 identified primer pairs, synthesized, and used to determine if the EST-SSR loci identified in this study were true-to-type EST-SSR loci in the sainfoin populations. Of the 200 primer pairs, 178 of them successfully amplified fragments from sainfoin genomic DNA during PCR, while the other 22 primer pairs failed. Also, 132 of the 178 PCR primer pairs amplified products of the expected size. Using genomic DNAs from the 40 different wild sainfoin plants (Table S3) as templates, 61 of the 132 primer pairs were found to be polymorphic (Fig. S6 and Table S4) and the other 71 primer pairs were monomorphic (Table S2).

## Assessment of sainfoin genetic diversity

The 61 primer pairs mentioned above were used to analyze the genetic diversity among the population comprising 40 wild sainfoin plants from five different geographic locations. The result showed that a total of 459 alleles were present in the 61 polymorphic loci in the 40 different individuals, and the number of alleles per loci ranged from three to twelve with an average number of 7.52. The Ho, He, and PIC were estimated from 0.05 to 1.0 (mean value = 0.67), 0.55 to 0.91 (mean value = 0.77), and 0.51 to 0.88 (mean value = 0.74), respectively (Table S4). These 61 polymorphic loci or EST-SSR markers are unlinked and

have high degrees of universality among the assayed germplasms. Therefore, they are useful for studying biogeographic processes that shaped the current disjunctive distributions of sainfoin. Furthermore, PCR amplicons representing EST-SSRs from different sainfoin individuals were sequenced and the results showed that all of the sequenced alleles were homologous to the original locus from which the marker was designed.

Using NTSYSPC 2.0 software and the UPGMA method, a dendrogram was obtained (Fig. 2). The 40 sainfoin individuals can be divided into five distinct wild populations. The individuals in population 1 originated from Minqin, the individuals in population 2 were from Jingyuan, the individuals in population 3 were from Yuzhong, the individuals in population 4 were from Huining, and the individuals in population 5 were from Maqu (Table S3). The individual plants from the Jingyuan sampling site were assessed on related taxa within the sainfoin genus. Most of them were easily amplifiable and detectable across all genotypes, and just a few showed problems of amplification or scoring, most likely due to polymorphisms (insertions/deletions or base mutations) in primer regions or in regard to subspecies. However, most EST-SSR markers found in this study are highly useful for discriminating related sainfoin populations or related taxa, including wild species, subspecies, and subgenera. Consequently, we propose that these EST-SSR markers can be used in future studies on comparative genomics, genetic differentiation, and evolutionary dynamics within the sainfoin genus.

STRUCTURE 2.3.4 was used to generate a structure map. STRUCTURE analysis based on 459 loci representing EST-SSRs was performed to evaluate the genetic structure of the 40 wild sainfoin individuals. The highest $\Delta K$ was observed for $K = 6$ [$\Delta K(6) = 144$]. $\Delta K$ values for $K = 3$–5 and $K = 7$ were not significant ($\Delta K = 0.078$–5.632). The mean value of the log probability of the data was higher for $K = 6$ than for $K = 4$, and $K = 5$ [LnP (D) $K = 6 = -13190.22$, LnP (D) $K = 4 = -14038.11$, LnP (D) $K = 5 = -13599$] (Table S6). Therefore, six clusters were chosen as the most probable genetic structure of the wild sainfoin individuals. With $K = 6$, seven individuals from site 1 were assigned to cluster 1 with coefficient $Q$ values ranging from 0.782 to 1.000; one individual from site 2 to cluster 2 with a $Q$ value between 0.900 and 1.000; one from site 2 to cluster 3 with a $Q$ value between 0.900 and 1.000; three from site 2 and three from site 3 to cluster 4 with $Q$ values from 0.782 and 0.982; seven from site 4 to cluster 5 with $Q$ values from 0.683 to 0.973; and eight from site 5 to cluster 6 with $Q$ values from 0.714 and 0.965 (Fig. 3). Ten sainfoin individuals could not be assigned to any of the clusters due to high levels of admixture ($Q < 0.6$).

Furthermore, PCR amplicons of Vo61 and Vo157 EST-SSRs from single individuals were sequenced to check the authenticity of the SSR locus (Fig. S6 and Fig. 4). For individual 2 in accession II at the Vo61 locus, four amplicons with different AGAA repeats, $(AGAA)_3$, $(AGAA)_4$, $(AGAA)_5$, $(AGAA)_6$, were found by pGEM-T easy vector sequencing, while three amplicons with different ACC repeats, $(ACC)_3$, $(ACC)_5$, $(ACC)_6$, were found for individual 1 in accession II for the Vo157 locus.

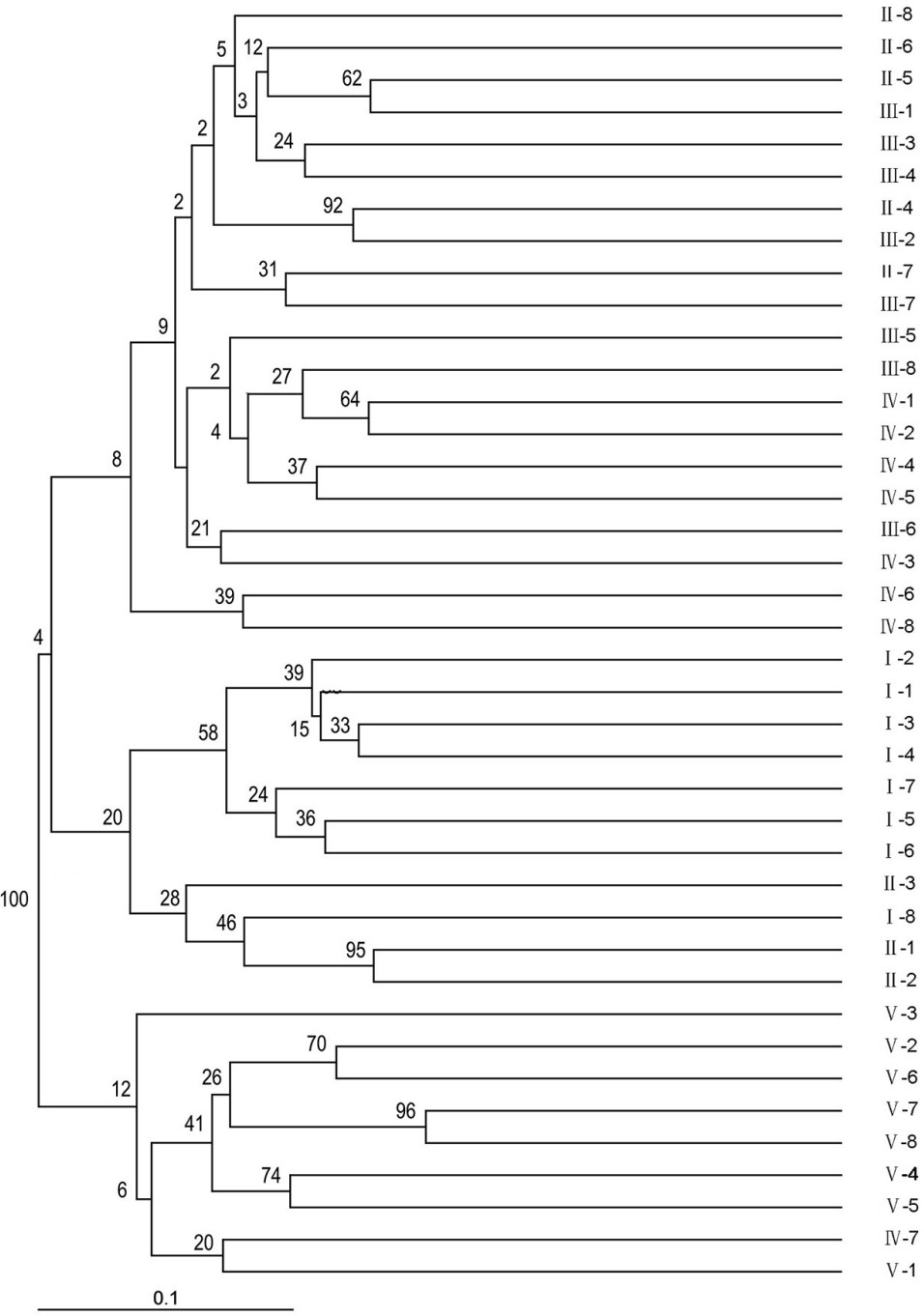

**Figure 2** **Phylogenic relationships among the 40 wild sainfoin individuals. The phylogeny tree was constructed using a neighbor-joining dendrogram in the Darwin software.** The starting dataset was represented by the 61 best EST-SSRs. I–V, group number representing five different sampling locations. 1–8, sample number representing eight individual samples in the same group.

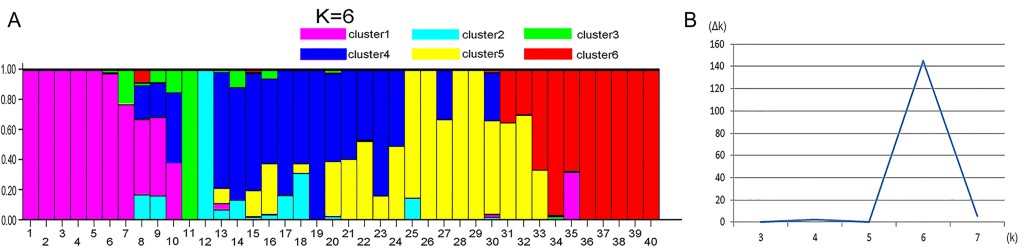

**Figure 3** **STRUCTURE analysis of the genetic structures of five vetch sainfoin.** Six different colors represent six different clusters. (A) Pink, cluster 1; green, cluster 2; cyan, cluster 3; blue, cluster 4; yellow, cluster 5; and red, cluster 6. (B) The value of the predicted population size ( $\Delta k$). Genetic structure of eight individuals in each of the five sainfoin sample populations is inferred by STRUCTURE using the EST-SSR markers dataset.

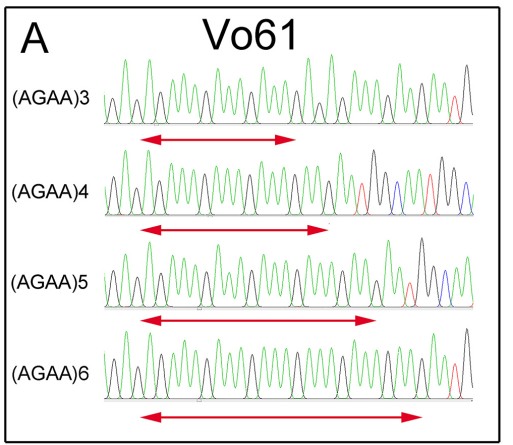
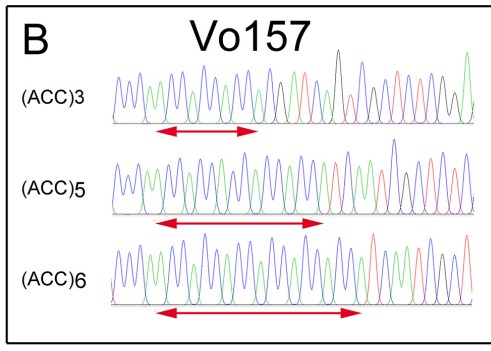

**Figure 4** **Comparative analysis of the DNA fragment peak spectrum for two selected EST-SSR loci among the same unit of *Onobrychis viciifolia*.** (A) Vo61 selected individual from II-2. (B) Vo157 selected individual from II-1.

## DISCUSSION

Transcriptome sequencing followed by *de novo* assembly was shown to be a useful tool for gene and molecular marker discovery for this study. Here, we showed that short reads obtained using Illumina paired-end sequencing of sainfoin cDNAs could be quickly assembled and used for transcriptome analysis, marker development and gene identification without a reference sainfoin genome. The results of our marker validation assay agreed with previous investigations on SSR markers of common bean (*Schmutz et al., 2014*) and SSR markers in other legume crops (*Kang et al., 2014*) where EST-SSR markers detect moderate polymorphism. *De novo* transcriptome sequencing was considered a crucial tool for gene function study and development of molecular markers (*Garg et al., 2011*; *Kaur et al., 2012*; *Duarte et al., 2014*; *Yates et al., 2014*). For legume plants, whole genome sequences of *Medicago* (*Young et al., 2011*), soybean (*Schmutz et al., 2010*), common bean (*Schmutz et al., 2014*), mung bean (*Kang et al., 2014*), and adzuki bean (*Kang et al., 2015*)

were reported. In this study, the assembled unigenes were analyzed by BLAST searching the available databases, and a total of 36,353 unigenes (47% of the assembled unigenes) were annotated. In addition, the identity of 46% of the assembled unigenes was obtained by BLASTX searching against the NR database, although this percentage was slightly lower than that reported for other plants, including orchid (49.25%) (*Zhang et al., 2013*), sesame (53.91%) (*Wei et al., 2011*), and litchi (59.65%) (*Li et al., 2013*). It is possible that the current incomplete sainfoin genomic and transcriptomic information affected our annotation efficiency and left some sainfoin-specific genes unidentified.

In this study, a total of 6,752 potential EST-SSRs were identified in the 77,764 unigenes. The frequency of EST-SSRs was one SSR per 4.35-kb sequence, which is much higher than what was reported for pineapple (1 in 13 kb) (*Ong, Voo & Kumar, 2012*), and lotus (1 in 13.04 kb) (*Pan et al., 2010*). However, this frequency is lower than what was reported for Levant cotton (1 in 2.4 kb) (*Jena et al., 2012*), castor bean (1 in 1.77 kb) (*Qiu et al., 2010*), and radish (1 in 3.45 kb) (*Wang et al., 2012*). It was speculated that the estimated frequency of SSRs depended strongly on the size of the database, SSR search criteria, and mining tools used (*Varshney, Graner & Sorrells, 2005*). In our study, mono-repeats were the most abundant repeat type. Our GO enrichment analysis showed that unigenes related to the category "transcription" were significantly enriched. This finding agreed with our previous investigations on alfalfa and *Vicia sativa* using similar GO analyses of SSR-containing unigene approaches (*Liu et al., 2013a*; *Liu et al., 2013b*). The results of GO enrichment analysis of unigenes containing EST-SSR loci indicated that EST-SSR loci are not randomly distributed in the transcriptome and it preference exists in transcription factors (Fig. S5C). These results are consistent with previous studies (*Luo et al., 2015*; *Zhou et al., 2016*).

The analysis of transcriptomes from 14 distinct sainfoin individuals using the Illumina HiSeqTM 2500 platform generated a total of 24.63 million clean reads, equivalent to a total of 6,264,706,761 bp length. Approximately 91.5% of the clean reads had Phred quality scores at the Q30 level and an N percentage (percentage of ambiguous "N" bases) of 0. The quality of the clean reads indicated a quality sequencing result. A total of 77,764 unigenes were assembled and had a mean unigene length of 682.01 bp. This mean length was greater than what was reported for tea (402 bp) (*Tan et al., 2013*) and sweet potato (581 bp) (*Wang et al., 2010*), possibly because the paired-end reads (100 bp) obtained in this study were longer than those (75 bp) used in previous studies (*Jia et al., 2015*). It is noteworthy that the 100-bp paired-end reads obtained in this study were shorter than what was documented in other reports, including alfalfa (803 bp) (*Liu et al., 2013a*) and seashore paspalum (970 bp) (*Jia et al., 2015*). Thus, the longer mean unigene length obtained in this study may also contribute to the different parameters used during sequence assembly and the nature of the plant. In addition, we thought that the Illumina sequencing technology used in this study also helped to allow better discoveries of novel unigenes and marker development for sainfoin. A neighbor-joining dendrogram based on allele distances showed the genetic relationships among the 40 sainfoin individuals (Fig. 3).

ATETRA 1.2.a software was used to calculate the number of alleles per locus, allele size, Ho, He, genetic distances, genetic similarity between individual sainfoins, and PIC defined as a closely related diversity measure. The average PIC value obtained in this study

was 0.74, which is higher than that (0.43) reported for sainfoin (*Mora-Ortiz et al., 2016*). The difference between the two PIC values might be caused by the different materials and different loci used in these two studies. For example, the EST-SSRs used in our study were from 14 different sainfoin tissues, but the markers used by Mora-Ortiz et al. were generated from 7-day-old seedlings. Our study focused on 40 wild sainfoin individuals, and Mora-Ortiz et al. used 32 sainfoin individuals representing distinct varieties or landraces. Additionally, the 61 highest polymorphic EST-SSR markers were selected from 200 EST-SSR primer pairs in our study.

The dendrogram and STRUCTURE map showed no clear relationship between the clustering pattern and geographical distance, which may be due to the lack of adequate accession numbers and the fact that these sainfoin accessions were sampled from adjacent areas, where the frequent exchange of sainfoin germplasm may obscure an existing pattern following the geographical origin of the accessions. In addition, this vague geographical pattern may be related to the autotetraploid characteristics of sainfoin. Therefore, the use of a higher number of accessions from close geographical locations and more individual plants per accession will be essential for verifying the genetic diversity of sainfoin in future studies.

In heterologous hexaploid wheat, there are one or two amplicons in a single individual at an SSR locus (*Yang, Peng & Yang, 2016*; *Sipahi et al., 2017*). The autotetraploid plants alfalfa (*Liu, Liu & Yang, 2007*) and potato (*Chandel et al., 2015*) show one, two, three or four amplicons in single individuals at an SSR locus. We found that the number of bands in sainfoin individuals ranges from one amplicon to four amplicons. For example, one band was found at Vo 61 in the No. 7 plant in accession II , two bands in the No. 4 plant, three bands in the No. 3 plant, and four bands in the No. 2 plant (Fig. S6). Additionally, by sequencing the four bands in the No. 2 plant, different AGAA repeats were found in the four amplicons $(AGAA)_3$, $(AGAA)_4$, $(AGAA)_5$, $(AGAA)_6$ (Fig. 4). In the previous study, there are some reports claimed that sainfoin is autotetraploid (*Hayot Carbonero et al., 2013*; *Mora-Ortiz et al., 2016*). The sequencing results of EST-SSR alleles in our study are consistent with the characteristics of autotetraploid. The results provided new evidence from EST-SSR molecular markers that sainfoin is autotetraploid.

## CONCLUSIONS

In this study, a total of 24,630,711 clean reads were generated from 14 different sainfoin tissue samples using Illumina paired-end sequencing technology. The reads were deposited into the NCBI SRA database (SRX3763386). From these clean reads, 77,764 unigene sequences were identified and resulted in 6,752 EST-SSRs. Using this information, 61 novel EST-SSR markers were developed for sainfoin and successfully used to confirm the genetic diversities among the 40 randomly collected wild sainfoin individuals, representing five different geographic regions. Additionally, sainfoin was re-verified to be autotetraploid by counting the EST-SSRs band number and sequencing the bands in one sainfoin individual. These 61 EST-SSR markers have relatively high degrees of polymorphism and can be used

in studies on genetic diversity, cultivar identification, sainfoin evolution, linkage mapping, comparative genomics, and/or marker-assisted selection breeding of sainfoin.

### Funding

This work was supported by the National Natural Science Foundation of China, Grant/Award Numbers: 31722055, 31672476, and 31730093; Fundamental Research Funds for the Central Universities, Grant/Award Numbers: lzujbky-2017-ot22 and lzujbky-2017-it08. The funders had no role in study design, data collection and analysis, decision to publish, or preparation of the manuscript.

### Grant Disclosures

The following grant information was disclosed by the authors:
National Natural Science Foundation of China: 31722055, 31672476, 31730093.
Fundamental Research Funds for the Central Universities: lzujbky-2017-ot22, lzujbky-2017-it08.

### Competing Interests

The authors declare there are no competing interests.

### Author Contributions

- Shuheng Shen performed the experiments, analyzed the data, contributed reagents/materials/analysis tools, prepared figures and/or tables, authored or reviewed drafts of the paper, approved the final draft.
- Xutian Chai performed the experiments, analyzed the data, prepared figures and/or tables.
- Qiang Zhou approved the final draft.
- Dong Luo analyzed the data.
- Yanrong Wang conceived and designed the experiments, contributed reagents/materials/analysis tools.
- Zhipeng Liu conceived and designed the experiments, contributed reagents/materials/analysis tools, prepared figures and/or tables, authored or reviewed drafts of the paper, approved the final draft.

### Data Availability

Data is available via the NCBI SRA database (SRX3763386).

### Supplemental Information

Supplemental information for this article can be found online at http://dx.doi.org/10.7717/peerj.6542#supplemental-information.

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
