# Peer review of "Development of polymorphic EST-SSR markers and characterization of the autotetraploid genome of sainfoin (Onobrychis viciifolia)"

_PeerJ, doi:10.7717/peerj.6542_

## Round 0.1 · original submission · Major Revisions

Thank you for submitting this paper to PeerJ for review. We've received comments from at least two reviewers, who have suggested a number of minor and major revisions (see below). Please submit your revisions or rebuttal to each of these suggestion for us to reconsider this paper.

·

Basic reporting

This article is mostly written in clear, unambiguous, professional English. Care should be given to "de novo" vs "denovo" used throughout the manuscript, referencing not consistent (et. al., 20xx vs et. al. 20xx), spacing between references (especially in Discussion) and other typing and spelling errors.

Literature are sited throughout. I would have preferred a bit more information on the uses of sainfoin, especially in animal husbandry and nutrition.

Figure 2 was not referenced in the text at all.

Reporting on the sequence, assembly and annotation of the transcriptome is sufficient and well written. The validation of the wild sainfoin populations as well as the verification of sainfoin autoteraploidy is inconsistent between materials and methods, results and discussion and need more attention.

A better distinction between di/tri/tetra (etc)-nucleotides and the number of repeats should be made as it gets confusing what the authors mean.

The application of the EST-SSRs, once identified, is not clear. The authors only mention "...should benefit sainfoin improvement projects..." (lines 86-87) but more information on applications will give a clearer picture of the future application of the project.
The authors mention the objective is to "allow a better understanding of the diversity of sainfoin" (line 92). However, with 40 individuals from only 5 locations this could hardly be achieved.

Experimental design

The experiment was well designed. The aim (transcriptome sequencing) and two objectives (diversity and autotetraploidy) were clearly defined. However, more than 40 individuals should be tested before final conclusions on genomic diversities can be made. The authors can elaborate on why the use of sainfoin plants in agriculture is important and why this species was selected for sequencing, as well as the down-stream application of the identified EST-SSRs.

The transcriptome sequencing, assembly and annotation of EST-SSRs were well reported in materials and methods as well as results and discussion.

Sampling of wild populations, DNA extraction and amplification were well reported. The construction of T4 vectors were very poorly described (especially sequencing) and need more attention. Diversity analysis of wild populations were insufficient and inconsistent between materials and methods, results and discussion. For example, in materials and methods it is noted that population parameters were analysed using ATETRA 1.2, and in Discussion the authors mention that POPGENE were used to analysed some and other population parameters. No reference of which program was used is made in Results. No mention is made of which sequencing method is used (e.g. Sanger or NGS) of the T4 vectors and methods and results can be elaborated. The authors should pay attention to these sections and ensure correct reporting throughout.

The results from Figure 4 should be elaborated and better explained in materials and methods, results and discussion.

Validity of the findings

The transcriptome sequencing and EST-SSR identification data is well written, statistically sound and controlled.

The diversity analysis data in lines 225 - 236 and 237 - 249 should perhaps be given in table format as it is difficult to read.

EST-SSR identification was conclusive and compared well with results from similar projects.

Line 145-146 states the plants had different tannin contents and compositions. No data is given to state these claims.

The legends of Figures 2 and 3 are swapped and no mention of Figure 2 or an interpretation of the data is given. The authors should elaborate much more on these findings.

More detail on where EST-SSRs are located in the transcriptome (lines 235-236) can be given to show that included EST-SSRs cover the whole transcriptome.

The results of the autotetraploidy analysis is lacking and more detail should be given here.

The diversity analysis should be given more attention and it should be justified why only 40 individuals were used.

Conclusion is well stated.

Additional comments

Overall not a bad article but it needs more attention to the following:
- why sainfoin is important especially in agriculture and why this study was needed;
- diversity and autotetraploidy verification sections need more detail and attention to detail;
- whole article need to be checked for spelling errors, detail typing errors and inconsistent referencing;
- Figure 2 not addressed and legends incorrect;
- Figure 2 and 3 should be elaborated on;
-Re-write lines 80 - 81 as it is not clear;
- Line 52 - give SSR abbreviation, line 57 - write out EST;
- Lines 67 - 69: why is the current EST-SSR primers insufficient?

·

Basic reporting

Your title needs to be more clear and specific to match the content. In line 2, this is my suggestion: "...characterization of the autotetraploid genome of sainfoin..."

Experimental design

Lines 96 to 100. Were these tissues harvested from different saifoin plants or was it from the same plant? The reader will need to know this information.
If from different plants, how many were they and was it the same species/ clone? This will help the reader determine if the variation is a result of genotypic differences or not

When you were doing the cDNA library construction, how many illumina lanes were run? This information will give an idea of the sequence depth

Line 144. These wild saifoin plants; were they different species from O. viviifolia? Please write the species name. Are they also tetraploid, or this information is not known at this moment?

Line 157. How were the 200 primer pairs identified? Tell this to your readers please

Validity of the findings

When you were doing rigorous quality checks on the sequence reads, what did this rigor involve? What was the requirement?

In your results concluding paragraph (line 267, 268), you concluded that sainfoin is autotetraploid. However, you only used the number of PCR amplicons to come to this conclusion. I think you need to explain how much more powerful this method is compared to flow cytometry which is the norm when doing ploidy analysis?

Additional comments

I think you have done a good job and that this research will be very useful to the scientific community. My only striking observation that I noticed is that there are slight details which you have and need to include. Overall, your research adds valuable knowledge to the academic and scientific community.

---

## Round 0.2 · accepted · Accept

The amendments were made as requested and the document is acceptable for publication.

# ·

Basic reporting

The authors addressed all my issues and I'm happy with the review.

Experimental design

The authors addressed all my issues and I'm happy with the review.

Validity of the findings

The authors addressed all my issues and I'm happy with the review.

Additional comments

I'm happy to accept this revised article.